# The impact of voluntary front-of-pack nutrition labelling on packaged food reformulation: A difference-in-differences analysis of the Australasian Health Star Rating scheme

**Laxman Bablani**[1]*, **Cliona Ni Mhurchu**[2,3,4], **Bruce Neal**[3,4], **Christopher L. Skeels**[5], **Kevin E. Staub**[5], **Tony Blakely**[1]

**1** Centre for Epidemiology and Biostatistics, Melbourne School of Population and Global Health, University of Melbourne, Melbourne, Australia, **2** Faculty of Medical and Health Sciences, University of Auckland, Auckland, New Zealand, **3** The George Institute for Global Health, Sydney, Australia, **4** University of New South Wales, Sydney, Australia, **5** Department of Economics, University of Melbourne, Melbourne, Australia

* bablanil@unimelb.edu.au

**Data Availability Statement:** Because of commercial and legal restrictions to the use of copyrighted material it is not possible to share data

## Abstract

### Background

Front-of-pack nutrition labelling (FoPL) of packaged foods can promote healthier diets. Australia and New Zealand (NZ) adopted the voluntary Health Star Rating (HSR) scheme in 2014. We studied the impact of voluntary adoption of HSR on food reformulation relative to unlabelled foods and examined differential impacts for more-versus-less healthy foods.

### Methods and findings

Annual nutrition information panel data were collected for nonseasonal packaged foods sold in major supermarkets in Auckland from 2013 to 2019 and in Sydney from 2014 to 2018. The analysis sample covered 58,905 unique products over 14 major food groups. We used a difference-in-differences design to estimate reformulation associated with HSR adoption.

Healthier products adopted HSR more than unhealthy products: >35% of products that achieved 4 or more stars displayed the label compared to <15% of products that achieved 2 stars or less. Products that adopted HSR were 6.5% and 10.7% more likely to increase their rating by $\geq 0.5$ stars in Australia and NZ, respectively. Labelled products showed a −4.0% [95% confidence interval (CI): −6.4% to −1.7%, $p = 0.001$] relative decline in sodium content in NZ, and there was a −1.4% [95% CI: −2.7% to −0.0%, $p = 0.045$] sodium change in Australia. HSR adoption was associated with a −2.3% [−3.7% to −0.9%, $p = 0.001$] change in sugar content in NZ and a statistically insignificant −1.1% [−2.3% to 0.1%, $p = 0.061$] difference in Australia. Initially unhealthy products showed larger reformulation effects when adopting HSR than healthier products. No evidence of a change in protein or saturated fat content was observed.

A limitation of our study is that results are not sales weighted. Thus, it is not able to assess changes in overall nutrient consumption that occur because of HSR-caused

openly which reveal the product or supermarket names, but unredacted versions of the dataset are available with a licensed agreement that they will be restricted to non-commercial use. For access to Nutritrack, please contact the The National Institute for Health Innovation at the University of Auckland at enquiries@nihi.auckland.ac.nz. For access to FoodSwitch, please contact Fraser Taylor, managing director for Foodswitch ftaylor@georgeinstitute.org.au or foodswitch@georgeinstitute.org.au.

**Funding:** T.B. and C.NM received funding for this study. This was an investigator-initiated study funded by a Health Research Council of New Zealand (http://www.hrc.govt.nz/) programme grant (18/672). The funders had no role in study design, data collection and analysis, decision to publish, or preparation of the manuscript.

**Competing interests:** I have read the journal's policy and the authors of this manuscript have the following competing interests: C.NM is a member of the New Zealand Health Star Rating Advisory Group. The Advisory Group had no role in the study design, data collection and analysis, decision to publish, or preparation of the manuscript.

**Abbreviations:** CEM, coarsened exact matching; CI, confidence interval; DALY, disability-adjusted life year; FoPL, front-of-pack nutrition labelling; FVNL, fruit, vegetable, nut, and legume; HSR, Health Star Rating; ID, identifier; NIP, nutrient information panel; NZ, New Zealand; SKU, stock keeping unit; STROBE, Strengthening the Reporting of Observational Studies in Epidemiology.

reformulation. Also, participation into labelling and reformulation is jointly determined by producers in this observational study, impacting its generalisability to settings with mandatory labelling.

## Conclusions

In this study, we observed that reformulation changes following voluntary HSR labelling are small, but greater for initially unhealthy products. Initially unhealthy foods were, however, less likely to adopt HSR. Our results, therefore, suggest that mandatory labelling has the greatest potential for improving the healthiness of packaged foods.

---

## Author summary

### Why was this study done?

- Front-of-pack nutrition labelling (FoPL) systems may lead to reformulation of food to healthier compositions.

- In December 2014, Australia and New Zealand (NZ) adopted the voluntary Health Star Rating (HSR) scheme on packaged food products.

- We studied whether the HSR was associated with industry-led reformulation.

- We also determined whether reformulation differed between products that were initially more healthy or unhealthy.

### What did the researchers do and find?

- Healthier products are more likely to show HSR scores than unhealthy ones: >35% of products that should have achieved 4 or more stars displayed the label compared to <15% of products that should achieve 2 stars or less.

- Compared to unlabelled products, products that adopt HSR are 6.5% and 10.7% more likely to increase their HSR by ≥0.5 stars in Australia and NZ, respectively. Labelled products showed a −4.0% [−6.4% to −1.7%] relative decline in sodium content in NZ and −1.4% [−2.7% to −0.0%] decline in Australia. HSR adoption was associated with a −2.3% [−3.7% to −1.0%] change in sugar content in NZ and a statistically insignificant −1.1% [−2.3% to 0.1%] difference in Australia.

- Initially unhealthy products that adopt HSR increase their rating by more than 0.1 stars. This effect becomes smaller the greater the initial healthiness of the product—a 1-star increase in initial healthiness reduces reformulation by around 0.04 stars.

- A limitation of our study is that results are not sales weighted. Thus, it is not able to assess changes in food consumption that occur because of HSR-caused reformulation. Also, the voluntary adoption of HSR along with the observational nature of our study may impact the generalisability of our results, e.g., to a setting where such labels were mandatory.

**What do these findings mean?**

- Overall, the introduction of HSR had a small effect on product reformulation.

- The voluntary nature of the HSR program lowers effectiveness because labels were mostly placed on already-healthy products.

- Our results suggest that HSR adoption by unhealthy products should be mandated by governments to maximise reformulation.

## Introduction

Population-based approaches are required to combat unhealthy diets, which have been linked to several noncommunicable diseases, including cardiovascular diseases, diabetes, and cancer [1]. Voluntary front-of-pack nutrition labelling (FoPL) is one such approach increasingly used on packaged foods to promote healthier diets. Such labels are designed to allow consumers to discern healthier items more effectively than the descriptive back-of-package labels [2]. Consumers are more likely to choose products they perceive to be healthier [3]. The expectation of labelling influencing consumer choice towards healthier products, and subsequently affecting industry profits, can encourage industry-led reformulation of packaged food products. This paper studies the Health Star Rating (HSR) label that was adopted by Australia and New Zealand (NZ) in December 2014 [4]. Since its introduction, HSR has seen steadily increasing acceptance and was displayed on about 23% of NZ products in 2019, and 31% of Australian products in 2018 (Fig A in S1 Text graphs the percentage of foods using HSR across years in Australia and NZ). HSR is an interpretive aid, allowing products to easily convey nutritional information to consumers. Underpinning HSR is a nutrient profile score [5] that summarises the density of 4 negative nutrients (energy, saturated fat, total sugar, and sodium) and 3 beneficial components (fibre; protein; and fruit, vegetable, nut, and legume [FVNL] content). These scores are used to award a health rating, displayed as a star graphic of half to 5 stars, in increments of 0.5 stars. The greater the number of stars, the healthier the product is overall. Such summary-graded FoPLs are also used elsewhere, e.g., the Nutri-Score system in Europe [6].

There is limited evidence on the effectiveness of FoPL policies on food reformulation, and what evidence exists traverses a variety of FoPL systems. A systematic review of the effect of FoPL on industry-led reformulation identified 13 studies [7]. However, these 13 studies differ in products affected (packaged groceries or fast food); mandatory versus voluntary label implementation; and whether they highlight beneficial or adverse nutrient profiles. Another key challenge in estimating the causal effect of HSR and other voluntary FoPL programs is the bias in participating products—healthier products are more likely to participate. Such confounding may thus affect estimates of the reformulation effects of voluntary FoPL unless change within the same products is tracked over time. Only a small number of studies have analysed the impact of FoPL on the reformulation of packaged supermarket foods. Ricciuto and colleagues [8] conducted a before–after study on the effect of mandatory Canadian *trans*-fat labelling on 18 margarine products, finding that 13 of them reduced *trans*-fat contents following the adoption of labelling. Vyth and colleagues [9] studied the impact of the voluntary Dutch Choices summary logo on a sample comprising 23.5% of labelled products and found that 168 labelled products were reformulated to lower sodium density and increase fibre content. It did not, however, control for reformulation trends among unlabelled products. This study also

compared the nutrient content of newly developed participating products to pre-intervention nutrient means, concluding that the logo was associated with healthier new product development. However, the voluntary participation of healthier products precludes causal interpretations. Analyses of Chile's food labelling laws found limited anticipatory reformulation [10], with a subsequent study [11] finding large changes in energy consumption arising from sugar-sweetened beverages (−11.9 kcal/capita/day [95% confidence interval (CI): −12.0 to −11.9, $p < 0.001$], but being unable to split out changes due to industry-led reformulation or consumer behaviour changes. The observational nature of policy-based studies necessitates methods that account for confounding to identify the causal effect of reformulation.

In NZ, the only published analysis of the effect of HSR on reformulation used a before–after design, with data from 2014 and 2016 [12]. It observed a small but statistically significant decrease in energy and sodium content of HSR-labelled products. It also found increases in fibre content. This study, however, did not control for broader reformulation trends that may have affected products in the absence of labelling, limiting the casual validity of the results. An Australian difference-in-differences analysis of energy density of HSR-labelled products in 2016, relative to 2013, finds a 0.6% reduction in energy density among 1,004 food products [13] (−7.11 kJ/100 g [95% CI: −14.2 to −0.1, $p = 0.04$]; other nutrients were not reported). Morrison and colleagues [14] used 2013 and 2016 Australian nutrient information panel (NIP) data on 100 children's products adopting the HSR to perform a before–after comparison, finding no statistically significant change in energy [CI/$p$-value not reported] and a decrease in sodium (−20 mg/100 g or ml [SD/CI not reported, $p = 0.01$]). Generally, the relatively low adoption rates of HSR in 2016 (between 5% and 10% of all packaged foods) may impact the reliability of these estimates.

This study, therefore, evaluated whether adoption of HSR may have led to packaged food reformulation using panel data techniques [15]. It also analysed differences in reformulation by baseline product healthiness. In the context of voluntary labelling, this is an important analysis to help improve the capacity for policies such as HSR to affect reformulation.

## Methods

### Study overview

The primary data sources on food formulation were mandatory back-of-pack NIP data from both Australia (2014 to 2018) and NZ (2013 to 2019) on nonseasonal packaged products sold at major supermarkets in both countries. Using a difference-in-differences design with product- and time-level fixed effects, we controlled for both within-product confounding and market-wide reformulation effects that may have biased the results of our study. The outcomes examined were changes in the density of the 4 targeted negative nutrients in HSR—energy, sugars, saturated fat, and sodium; and 2 positive constituents—protein and fibre. Another outcome was the HSR rating score itself, which we imputed for all products at all time points using NIP data and the publicly available HSR algorithm. Lastly, we examined heterogeneous effects of HSR adoption by the healthiness of products defined using their imputed HSR ratings, before HSR label adoption.

Although a prospective design for this study does not exist, relevant portions of the grant funding for this study are provided in S9 Text. The grant specified using fixed-effects and difference-in-differences methods to study the impact of HSR on product reformulation, and this study conforms to the broad research design and questions therein. We note the following key changes from the grant: First, data available in late 2019 were used to provide timely evidence for the program. Second, the nutrient profile score was replaced with HSR score as an outcome to enhance the study's relevance, since most stakeholders only observe the HSR score. Data-

driven changes to the analysis include dropping a detailed analysis of FVNL composition across products due to proprietary algorithms used in imputing FVNL content. We also became aware of issues with fibre content within the datasets and ran analyses for robustness, as described below. Lastly, our reviewers provided many valuable suggestions for analyses to improve the clarity of our data sample. These include the addition of all analyses in S1 and S2 Text, as well as the refinement of the CEM weights used in S5 Text to include food group information.

Reporting of this study conforms to the Strengthening the Reporting of Observational Studies in Epidemiology (STROBE) guidelines [16] (S8 Text).

## Data sources

Data on the nutritional composition of nonseasonal packaged food products (stock keeping units [SKUs]) sold across several Australasian supermarkets were taken from 2 sources: Nutritrack 2013 to 2019 data for NZ, collected by the National Institute of Health Innovation at the University of Auckland [17] and FoodSwitch 2014 to 2018 data for Australia, collected by The George Institute for Global Health [18]. In both countries, the supermarket chains surveyed dominate packaged food distribution, accounting for at least 70% of overall grocery retail value in 2019 [19]. Over 2013 to 2019, the Nutritrack products surveyed accounted for 81.5% of purchases recorded in HomeScan NZ, a large consumer survey (excluding fresh food and alcohol purchases). Consumer panel information for Australia was not available. It was not possible to merge the NZ and Australian data.

Nutritrack collects information between February and April each year on all packaged food and beverages sold at 4 major supermarkets in Auckland, NZ; 1 store each of New World, 4Square, Countdown, and PAK'nSAVE chains. Key exclusions from the dataset include products that do not display a NIP, unpackaged fresh foods, bulk bin items, alcohol, seasonal products (such as Easter eggs), and dietary supplements.

Similarly, FoodSwitch collects annual data of product information from 4 stores in Sydney, Australia; 1 each of ALDI, Coles, IGA, and Woolworths chains. This field survey data are augmented by supplementary data collection and crowdsourcing through the FoodSwitch mobile app, which has been downloaded over 600,000 times.

Both photographic surveys present largely comparable information on packaged food products sold in each country. Each contains SKU codes, brand and product identifiers, and uses a unified food group coding system. They also contain data on all nutrients mandatorily listed on the NIP—energy, sodium, sugar, saturated fat, and protein. Additional nutrients and micronutrients, such as fibre, vitamins, or minerals, are also captured if listed on the NIP. The presence of FoPL, such as HSR and the actual HSR score, is contained within each dataset. Each dataset also performs an imputation of HSR across products, using ingredient information to calculate the FVNL content scores.

However, Nutritrack contains information on some food groups not in FoodSwitch—notably, eggs. Both datasets differently treat NIPs for foods that require preparation (say, dry soup mix); within each dataset, such NIPs are treated consistently. Nutritrack reports "as-sold" NIPs by default; FoodSwitch reports "as-prepared" NIPs. The small number of products affected (<5%), consistent treatment of NIPs within datasets, and use of fixed-effects methods ameliorate much of the impact of such differences. S2 Text provides more detail on the handling of "as-prepared" NIPs.

## Sample selection

Products not in the scope of HSR, such as alcohol, vitamins, supplements, and special foods (e.g., toddler food), were excluded from the analysis.

A format variant of the HSR, the energy-only icon, summarises energy per pack/serving but does not display a star rating and is used as an informational aid primarily on confectionary and nondairy beverage products. Due to differences in labelling style (informational versus interpretative) and lack of comparability with other HSR-labelled products, products with the HSR energy-only icon logos were excluded.

## Exposure and outcome variables

SKU-specific barcodes were used to link product information over time, forming 2 longitudinal country-specific datasets. Information on the display of HSR labels on a packet was used as the primary exposure variable.

Nutrient density and imputed HSR scores formed the outcome variables. Fibre was not mandatorily displayed on NIPs and was shown on roughly 40% of all observations. In some cases, fibre values were entered as 0 when they were missing. In such instances, these 0s were changed to missing. To maximise overall sample sizes, missing fibre values were imputed with leads and lags, i.e., if there was no evidence of product reformulation on other nutrients, then data on a missing fibre in 1 year were imputed with the value from an adjacent year. Sensitivity analyses for fibre content are presented in S6 Text—fibre results for the total (i.e., including imputed) data are attenuated compared to estimates without such imputations but gain precision. The FVNL content in each dataset was used to impute HSR ratings but was not chosen as an independent study outcome.

Using the nutrient information above and the publicly available HSR algorithm [20], an estimated HSR rating was imputed for all products at all time points within the sample. A comparison of imputed and displayed HSRs (calculated by the food producer) indicates that the imputations matched the displayed rating exactly for 76% of products, and 95% were within 1 star. For the remaining 4%, the imputed HSR is generally more than 1 star below displayed ratings. Many of these non-agreeing products reported "as prepared" NIP values, which are more nutritious than "as sold" NIP values, e.g., stir fry sauces and meal mixes. To enable a consistent before–after analysis, all analyses on HSR ratings used imputed HSR scores, and the "actual" displayed HSR was not used.

## Analysis

We used a difference-in-differences design, based on a before–after difference in levels for the outcome variable, in the presence of a comparison group to control for factors that would have affected the treatment group in the absence of treatment. The addition of a comparison group offers protection against time-varying confounding that may affect previous before–after studies. Here, the treatment group consisted of all products adopting HSR labelling during the sample period. The comparison group was products that never adopted HSR ratings throughout the study period. The regression model estimated was

$$y_{pt} = \alpha + \gamma_p + \mu_t + \beta \times hsr_{pt} + \epsilon_{pt}, \tag{Eq1}$$

where $y_{pt}$ was the outcome: the HSR rating or nutrient levels in product $p$ at time $t$. $\gamma_p$ was the product fixed effect, which controlled for observed or unobserved time-invariant confounding such as type of food product and average manufacturer characteristics. $\mu_t$ controlled for trends that affect all products in the datasets equally, such as countrywide trends of product reformulation. Finally, $\beta$ captured the effect of HSR adoption, where $hsr_{pt}$ is a dichotomous variable, coded "1" when the product $p$ displayed HSR at time $t$ and "0" otherwise. The error term $\epsilon_{pt}$ encapsulated factors other than $hsr_{pt}$, $\gamma_p$, or $\mu_t$ that affect the nutrient composition of products. Errors were clustered at the product identifier (ID) level.

All analyses were performed using Stata statistical software (Stata, Texas, United States of America).

## Stratification by pre-labelling imputed HSR rating

Products were split into 3 categories based on their baseline imputed HSR rating (before any adopted HSR labelling): 0.5 to 1.5, 2.0 to 3.5, and 4.0 to 5.0 stars. Category indicators were interacted with HSR participation in Eq 1 to estimate differential reformulation effects by pre-intervention HSR.

We also conducted formal tests for variation in reformulation by baseline HSR rating. First, the following variation of Eq 1. was estimated:

$$y_{pt} = \alpha + \gamma_p + \mu_t + \beta_0 \times hsr_{pt} + \beta_1 \times hsr_{pt} \times (baseline\ HSR\ rating_p - 0.5) + \epsilon_{pt}. \quad \text{(Eq2)}$$

$\beta_0$ is the average reformulation for an HSR product with the unhealthiest rating, 0.5. $\beta_1$ captured the change in the reformulation, over $\beta_0$, for each additional star in the HSR rating. A statistically significant coefficient for $\beta_1$ therefore provided evidence for differences in reformulation by baseline healthiness of foods.

Additionally, $t$ tests were conducted on the differences between the healthiest (4.0 to 5.0 stars) and remaining HSR categories (0.5 to 1.5 and 2.0 to 3.5) as another test for differences in reformulation.

## Sensitivity analyses

We conducted several robustness checks. Our results can be interpreted as causal if the "parallel paths assumption" is correct, namely that in the counterfactual absence of labelling for HSR-labelled food, changes in average reformulation over time are equal to those of unlabelled food. Although this assumption cannot be proved, we checked for the presence of differences in reformulation for HSR-labelled products before they underwent labelling, relative to unlabelled foods. The absence of significant differences strengthens the likelihood that the parallel paths assumption holds.

We also further controlled for time-varying confounders. Labelling with HSR is systematically linked to product nutrient composition. Such confounding, if varying with time, may bias the results of our study. We run 3 robustness checks to ameliorate threats from time-varying confounding. First, coarsened exact matching (CEM) is a nonparametric matching technique that balances pre-labelling nutrient and major food group information between HSR products and products that never received HSR labelling [21]. CEM-generated matching weights were used in Eq 1 to reduce bias associated with pre-labelling nutrition. Second, we controlled for linear preexisting trend differences between HSR-labelled and unlabelled foods. This allowed us to check our results when the parallel paths assumption required for difference-in-differences is violated and, therefore, acted as an important specification check [22]. Lastly, we combined CEM and differential trend approaches.

## Results

Table 1 gives the nutritional composition at baseline for products that did not adopt HSR (column 1) and products that adopted HSR labelling, in the year before labelling (column 2, treated group). These 2 groups comprise approximately 87,339 observations in NZ and 64,392 observations in Australia, mostly for unlabelled products. The reformulation effect for HSR-labelled products is based upon changes for products in the labelled group: 1,785 NZ and 2,462 Australian products. Products that took up HSR were healthier at baseline across all nutritional

 

**Table 1. Baseline product nutritional composition for the products that never adopted HSR (comparison group) and for those that did (treatment group) over the study period.**

| | NZ (Nutritrack 2013–2019) | | | | Australia (FoodSwitch 2014–2018) | | | |
|---|---|---|---|---|---|---|---|---|
| | Always unlabelled | Baseline for HSR labelled | Δ = Unlabelled −labelled | Total average | Always unlabelled | Baseline for HSR labelled | Δ = Unlabelled −labelled | Total average |
| Imputed HSR rating | 2.6 | 3.3 | −0.7 | 2.6 | 2.4 | 3.0 | −0.6 | 2.5 |
| | (2.5, 2.6) | (3.3, 3.3) | (−0.8, −0.7) | (2.6, 2.6) | (2.4, 2.4) | (3.0, 3.1) | (−0.6, −0.6) | (2.5, 2.5) |
| Energy kJ per 100 g/ml | 1,117.5 | 967.5 | 150.0 | 1,106.0 | 1,129.0 | 1,009.6 | 119.5 | 1,118.6 |
| | (1,112.1, 1,122.9) | (949.6, 985.4) | (130.6, 169.5) | (1,100.8, 1,111.2) | (1,122.8, 1,135.4) | (989.7, 1,029.6) | (98.2, 140.8) | (1,112.6, 1,124.6) |
| Sodium mg per 100 g/ml | 558.9 | 398.1 | 160.8 | 546.5 | 540.4 | 347.16 | 193.2 | 523.5 |
| | (544.9, 572.9) | (383.4, 412.7) | (112.3, 209.4) | (533.5, 559.4) | (525.7, 555) | (332.6, 361.7) | (145.2, 241.2) | (509.9, 537.1) |
| Sugar g per 100 g/ml | 14.2 | 9.2 | 5.0 | 13.8 | 14.3 | 9.2 | 5.1 | 13.9 |
| | (14.0, 14.3) | (8.8, 9.5) | (4.5, 5.5) | (13.7, 13.9) | (14.2, 14.5) | (8.8, 9.6) | (4.6, 5.7) | (13.7, 14.0) |
| Protein g per 100 g/ml | 7.1 | 7.6 | (0.5) | 7.1 | 7.2 | 7.6 | −0.4 | 7.2 |
| | (7.0, 7.2) | (7.4, 7.7) | (−0.7, −0.3) | (7.1, 7.2) | (7.1, 7.2) | (7.4, 7.8) | (−0.6, −0.2) | (7.2, 7.3) |
| Saturated fat g per 100 g/ml | 5.3 | 3.2 | 2.1 | 5.1 | 5.3 | 3.6 | 1.7 | 5.2 |
| | (5.2, 5.4) | (3.0, 3.3) | (1.9, 2.3) | (5.1, 5.2) | (5.3, 5.4) | (3.5, 3.8) | (1.5, 1.9) | (5.1, 5.3) |
| Fibre g per 100 g/ml | 1.3 | 2.2 | −0.9 | 1.3 | 1.2 | 2.6 | −1.4 | 1.4 |
| | (1.2, 1.3) | (2.1, 2.2) | (−1.0, −0.8) | (1.3, 1.4) | (1.2, 1.3) | (2.5, 2.7) | (−1.5, −1.3) | (1.3, 1.4) |
| Unique products | 28,053 | 1,785 | | 29,838 | 26,605 | 2,462 | | 29,067 |
| Observations | 80,694 | 6,645 | | 87,339 | 58,770 | 5,622 | | 64,392 |

Note: 95% CI in brackets.

CI, confidence interval; HSR, Health Star Rating; NZ, New Zealand.

measures, with higher imputed HSR ratings, fibre, and protein content and lower energy, sodium, sugar, and saturated fat content. Several additional descriptive analyses were also conducted. Table A in S1 Text presents the number of observations by food groups for products that never adopted HSR and those that adopted HSR across our study period and the last year of observation (2018 for Australia and 2019 for NZ). It finds that HSR adoption in both countries is led by cereals, convenience foods, processed meat, fish, fruit, and vegetable products. Lastly, Fig B in S1 Text graphs the overall trends in nutrient composition across the datasets in the study period showing, for instance, the energy density in the NZ sample increases from 1,095 to 1,134 kJ/100 g or ml, whereas the energy density of the Australian sample decreases slightly from 1,117 to 1,104 kJ/100 g or ml. Such underlying trends in overall sample composition highlight the reasons for using year and product fixed effects in our analysis, as they may confound analyses for the causal effect of HSR.

Fig 1 displays patterns of differential participation into HSR labelling for Australia (in 2018) and NZ (in 2019). In both countries, greater than 35% of products with an HSR score of 4.0 stars or more displayed an HSR label, compared to less than 15% of products with an expected HSR less than 2.

Fig 2 illustrates the percentage of products by imputed changes in HSR at the last year of the analysis period (2019 for NZ and 2018 for Australia), compared to before HSR was adopted (2013 to 2014). Each plot summarises data on the subsample of products observed both in 2013 to 2014 and 2018 to 2019, for approximately 6,000 products in each country. In both countries, over 70% of both HSR-labelled and unlabelled products did not see a change in HSR rating ($\Delta HSR = 0$). However, this proportion was lower for HSR-labelled products. Further, a clear skew towards more positive rating changes was seen for HSR-labelled products.

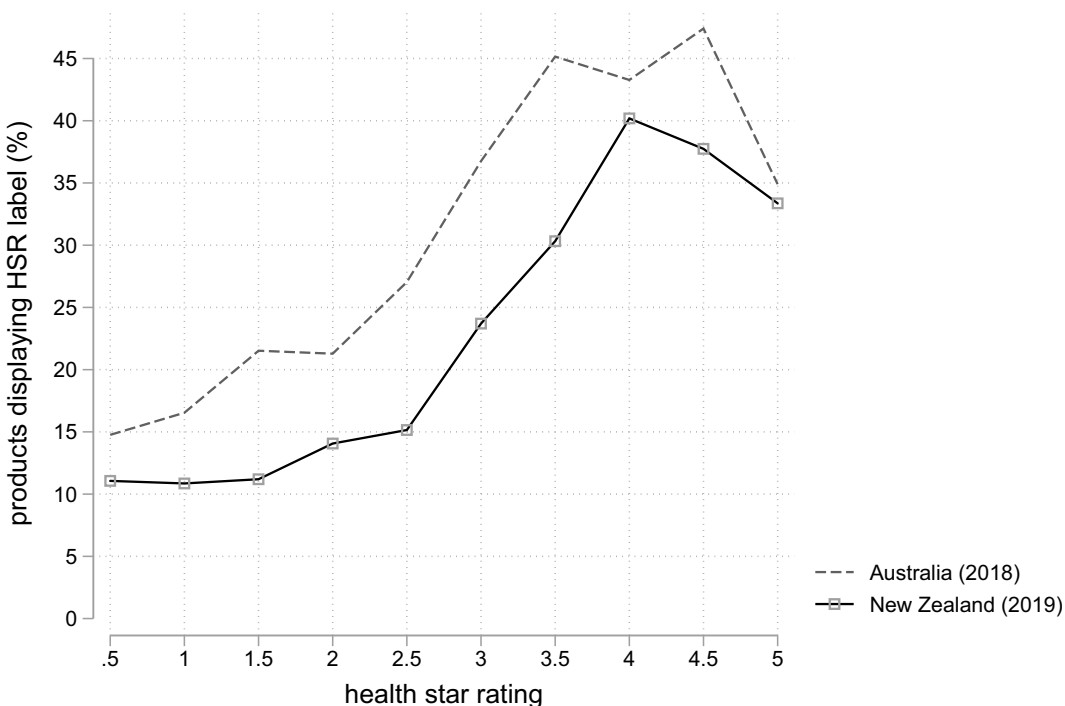

**Fig 1. Percent products displaying HSR labels by imputed HSR rating for Australia in 2018 and NZ products in 2019.** HSR, Health Star Rating; NZ, New Zealand.

Of note, 17.9% of Australian products saw HSR ratings increase by $\geq$ 0.5 stars, compared to 11.4% of unlabelled products, the difference between groups equalling 6.5%. NZ products saw 21.2% of HSR-labelled products increase by $\geq$0.5 stars, compared to 10.5% of unlabelled products, the difference between groups equalling 10.7%. Although Fig 2 did not contain data for

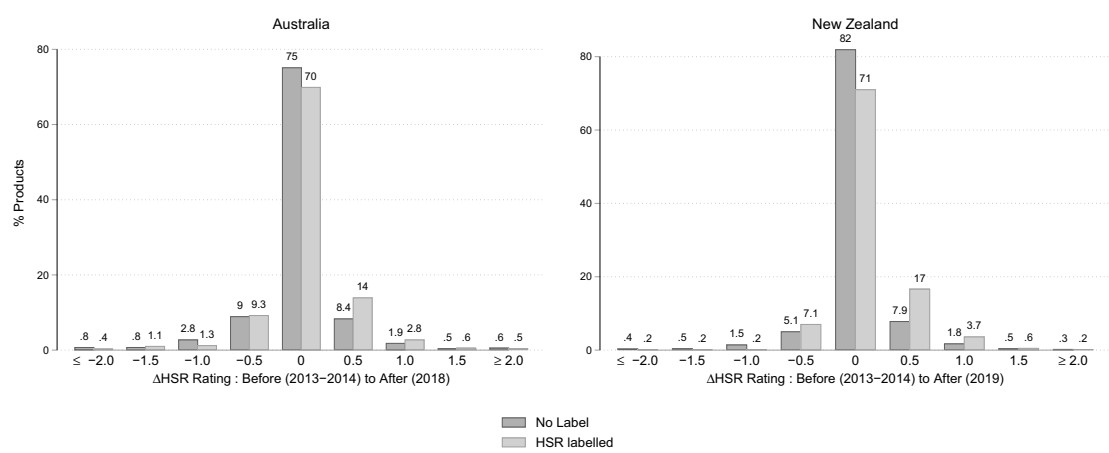

**Fig 2. Percent of products by change in imputed HSR rating before HSR implementation (2013–2014) versus the last year observed after adoption (2018 for Australia and 2019 for NZ).** HSR, Health Star Rating; NZ, New Zealand.

**Table 2. Effect of voluntary adoption of HSR label on nutrient profile score and component nutrients controlling for market-wide trends and time-invariant product-level characteristics using fixed-effects analysis.**

| | HSR rating | Energy (kJ per 100 g/ml) | Sodium (mg per 100 g/ml) | Sugar (g per 100 g/ml) | Protein (g per 100 g/ml) | Saturated fat (g per 100 g/ml) | Fibre (g per 100 g/ml) |
|---|---|---|---|---|---|---|---|
| NZ (Nutritrack 2013–2019) | | | | | | | |
| Absolute reformulation | 0.07 | −0.6 | −16.1 | −0.2 | −0.02 | −0.04 | 0.04 |
| | [0.05, 0.08] | [−5.1, 3.8] | [−25.3, −6.8] | [−0.3, −0.1] | [−0.08, 0.04] | [−0.09, 0.01] | [0.01, 0.08] |
| Relative reformulation (as % of pretreatment means) | 1.8 | −0.1 | −4.0 | −2.3 | −0.3 | −1.2 | 1.9 |
| | [1.4, 2.3] | [−0.5, 0.4] | [−6.4, −1.7] | [−3.7, −0.9] | [−1.0, 0.5] | [−2.9, 0.4] | [0.2, 3.6] |
| Observations | 90,088 | 93,377 | 92,894 | 92,633 | 93,372 | 93,348 | 95,239 |
| Australia (FoodSwitch 2014–2018) | | | | | | | |
| Absolute reformulation | 0.03 | −4.6 | −4.7 | −0.1 | −0.01 | 0.03 | −0.04 |
| | [0.02, 0.04] | [−7.9, −1.3] | [−9.4, −0.1] | [−0.2, 0.0] | [−0.05, 0.03] | [−0.01, 0.07] | [−0.08, 0.00] |
| Relative reformulation (as % of pretreatment means) | 1.0 | −0.5 | −1.4 | −1.1 | −0.1 | 0.8 | −1.6 |
| | [0.6, 1.5] | [−0.8, −0.1] | [−2.7, 0.0] | [−2.3, 0.1] | [−0.7, 0.4] | [−0.4, 2.0] | [−3.1, 0.0] |
| Observations | 77,797 | 78,232 | 78,339 | 78,149 | 78,200 | 77,934 | 78,339 |

Note: 95% CI in brackets.

CI, confidence interval; HSR, Health Star Rating; NZ, New Zealand.

all products underlying our fixed-effects analyses, it presents important descriptive evidence on the underlying pattern of reformulation.

Table 2 presents the fixed-effects analyses. In NZ, the HSR rating increased by 0.07 stars for labelled versus unlabelled foods, indicating that products that voluntarily adopt HSR labels reformulated to be healthier. In relative terms, this was a 1.84% (95% CI: 1.41 to 2.27, $p < 0.001$) increase in the number of stars for foods adopting the HSR label. Australian products showed a smaller increase of 0.03 stars or 1.01% (0.57% to 1.46%, $p < 0.001$). As indicated above, this was despite the majority of labelled products not changing HSR ratings.

The highest reformulation was found for sodium, which declined by −4.0% [95% CI: −6.4% to −1.7%, $p = 0.001$] in NZ, and −1.4% [−2.7% to 0.0%, $p = 0.045$] in Australia. Sugar content also decreased by more than 1%, declining by −2.3% in NZ [−3.7% to −0.9%, $p = 0.001$] and −1.1% in Australia [−2.3% to 0.1%, $p = 0.061$], although the reduction in Australia was not statistically significant. In contrast, energy density reductions were modest to negligible: −0.6 kJ/100 [95% CI: −5.1 to 3.8, $p = 0.773$] or −0.1% and not statistically significant in NZ and −4.6 kJ/100 [95% CI: −7.9 to −1.3, $p = 0.006$] or −0.5% but statistically significant in Australia. CIs for the decline in energy overlap in both countries, although we did not formally test for differences in reformulation in the 2 countries. Lastly, fibre saw a statistically significant increase of 1.9% [95% CI: 0.2% to 3.6%, $p = 0.027$] in NZ products compared to a decrease of −1.6% [−3.1% to −0.0%, $p = 0.047$] in Australia.

## Stratification by pretreatment imputed HSR rating

Fig 3 shows the reformulation effect on the star rating, stratified by the baseline HSR score. Reformulation was generally lowest for the healthiest products (4.0 to 5.0) in both countries. Australian products saw HSR scores decline slightly by −0.03 stars [95% CI: −0.06 to −0.01, $p = 0.001$] in this category, whereas a negligible change was observed in NZ. In contrast, reformulation was higher for the least healthy products (0.5 to 1.5). For this category, the HSR score increased by roughly 0.1 stars in both countries.

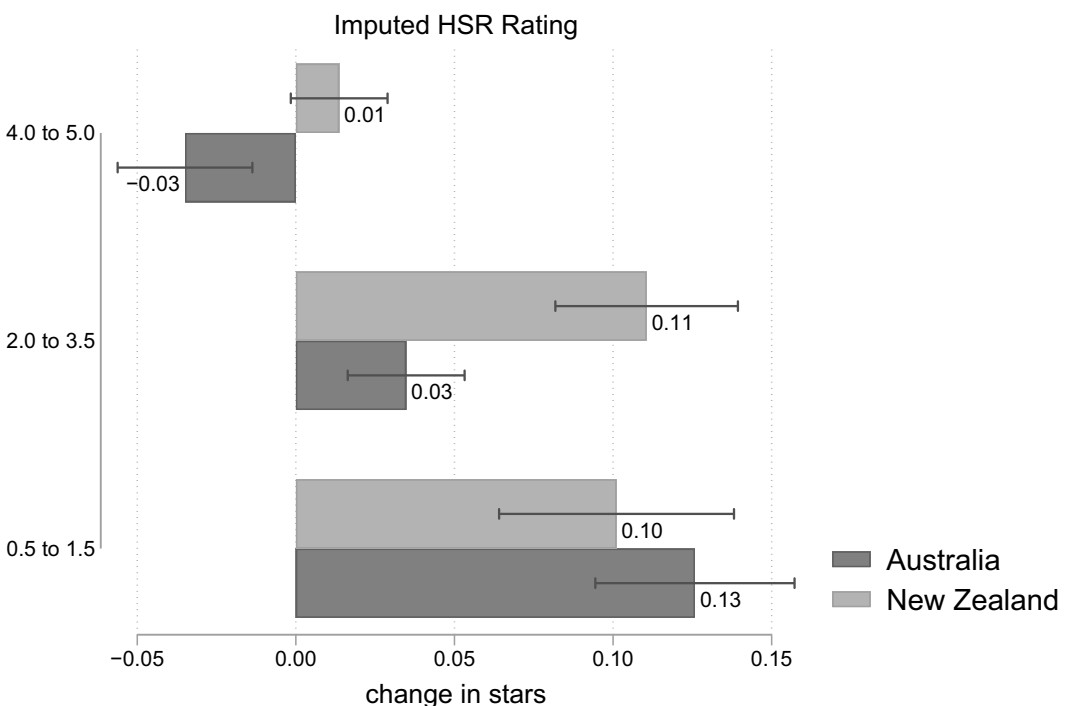

**Fig 3. Stratified analyses on the change in imputed HSR rating by pretreatment HSR—4.0 to 5.0 (most nutritious), 2.0 to 3.5, and 0.5 to 1.5 (least nutritious).** Regression estimates and 95% CIs displayed. Data used to construct this figure are in S3 Text. CI, confidence interval; HSR, Health Star Rating.

We also conducted analyses on the underlying nutrient densities. These results are displayed in Fig 4. For sodium content, reductions in NZ were roughly 15 times greater for the 0.5 to 1.5 HSR category compared with the 4.0 to 5.0 category (−59.35 mg versus −3.72 mg), and this pattern was also noted for Australia (−12.16 versus +1.57). Likewise, sugar content reductions were greater for the least healthy products in NZ (−0.46 versus −0.07) and in Australia (−0.72 versus +0.08). A similar pattern was noted for energy density (NZ: −2.15 versus +3.98; AU −13.74 versus +0.98), and saturated fat content (NZ: −0.09 versus +0.03; AU: −0.20 versus +0.11) in the 2 countries. Some evidence of increases in fibre content for unhealthy products was also observed in Australia, and no evidence of a change in protein content was found across either country.

Table 3 presents the results for our test of differences in reformulation by baseline HSR. The first row presents the estimated reformulation for products with baseline HSR = 0.5. The second row contains the change in reformulation when baseline HSR increases by 1. To illustrate, NZ products with baseline HSR = 0.5 showed a 0.17 increase in their star rating upon adoption of the HSR label (relative to non-adopting products), but this effect reduced by 0.04 for a 1-star higher rating. A product with a baseline HSR of 1.5 then saw its rating increase by 0.13 (or 0.17−0.04).

The results in the first row of Table 3 were greater than those in Table 2 across all outcomes, consistent with the hypothesis that the unhealthiest products underwent more reformulation than products on average. The second row estimates show that higher HSRs were generally associated with less reformulation than lower HSRs. For instance, a 1-star higher baseline HSR

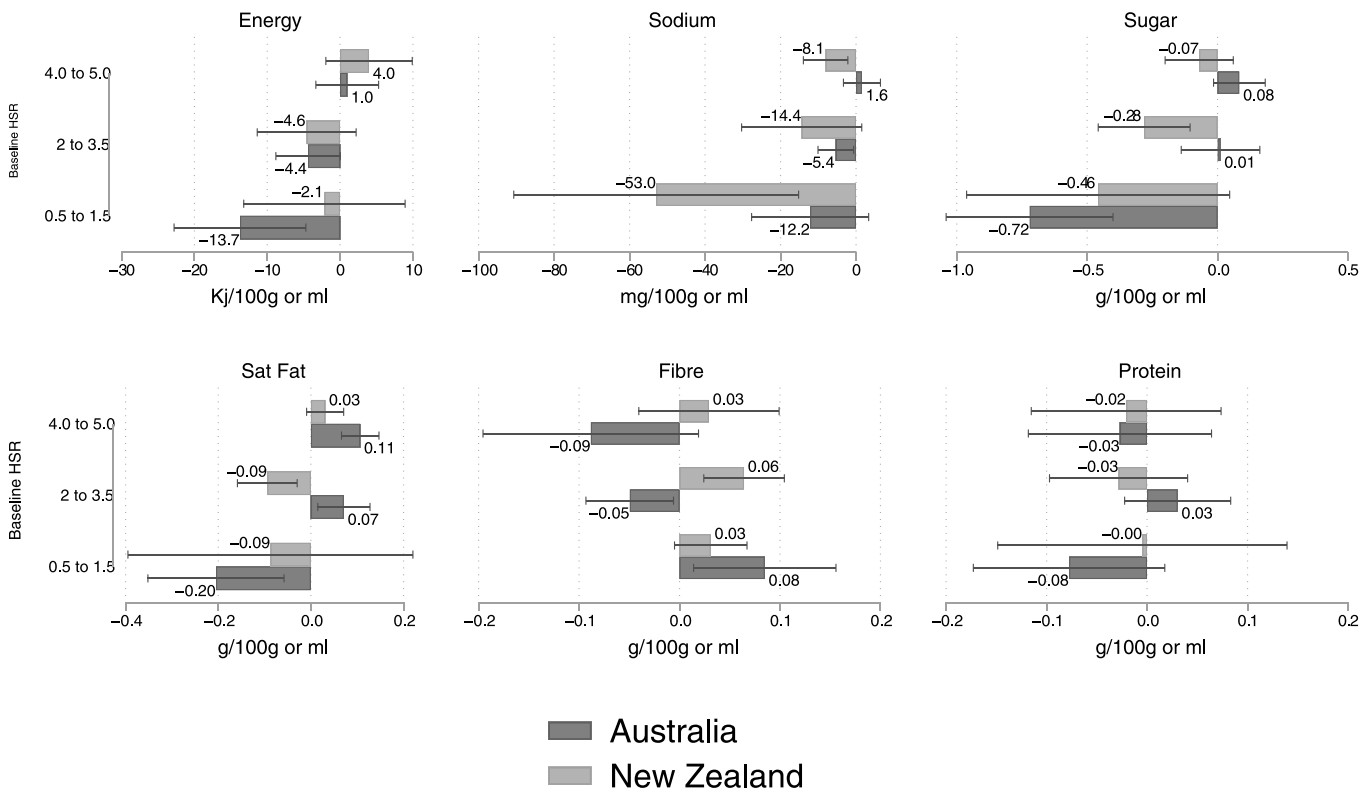

**Fig 4. Stratified analyses on the 6 nutrient density outcomes by pretreatment HSR—4.0 to 5.0 (most nutritious), 2.0 to 3.5, and 0.5 to 1.5 (least nutritious).**
Regression estimates and 95% CIs displayed. Data used to construct this figure are in S3 Text. CI, confidence interval; HSR, Health Star Rating.

**Table 3. Changes in reformulation following voluntary adoption of HSR by baseline HSR rating.**

| | HSR rating | Energy (kJ per 100 g/ml) | Sodium (mg per 100 g/ml) | Sugar (g per 100 g/ml) | Protein (g per 100 g/ml) | Saturated fat (g per 100 g/ml) | Fibre (g per 100 g/ml) |
|---|---|---|---|---|---|---|---|
| NZ (Nutritrack 2013–2019) | | | | | | | |
| Reformulation for baseline HSR = 0.5 | 0.17 | −6.5 | −42.0 | −0.73 | −0.01 | −0.16 | 0.10 |
| | [0.13, 0.21] | [−18.2, 5.2] | [−80.3, −3.5] | [−1.16, −0.29] | [−0.16, 0.13] | [−0.43, 0.11] | [0.02, 0.17] |
| Δ Reformulation for Δ baseline HSR = 1 | −0.04 | 2.1 | 8.9 | 0.18 | −0.00 | 0.04 | −0.02 |
| | [−0.05, −0.02] | [−1.5, 5.7] | [−2.0, 19.7] | [0.05, 0.31] | [−0.05, 0.05] | [−0.04, 0.12] | [−0.05, 0.01] |
| N | 86,209 | 87,429 | 87,025 | 87,020 | 87,400 | 88,231 | 87,429 |
| Australia (FoodSwitch 2014–2018) | | | | | | | |
| Reformulation for baseline HSR = 0.5 | 0.15 | −18.1 | −15.3 | −0.72 | −0.06 | −0.18 | 0.05 |
| | [0.12, 0.18] | [−26.9, −9.4] | [−28.6, −2.0] | [−1.03, −0.41] | [−0.16, 0.03] | [−0.32, −0.04] | [−0.02, 0.12] |
| Δ Reformulation for Δ baseline HSR = 1 | −0.05 | 5.3 | 4.3 | 0.25 | 0.02 | 0.08 | −0.03 |
| | [−0.06, −0.04] | [2.6, 8.1] | [0.2, 8.3] | [0.15, 0.34] | [−0.01, 0.06] | [0.04, 0.13] | [−0.07, −0.00] |
| N | 70,681 | 70,797 | 70,850 | 70,788 | 70,741 | 70,850 | 70,782 |

All estimates control for market-wide trends and time-invariant product-level characteristics using fixed-effects analysis.

Note: 95% CI in brackets.

CI, confidence interval; HSR, Health Star Rating; NZ, New Zealand.

reduced reformulation of sugar by 0.18 g/100 g [95% CI: 0.05 to 0.31, $p = 0.005$] in NZ and 0.25 g/100 g [0.15 to 0.34, $p < 0.001$] in Australia compared to the unhealthiest baseline HSR.

Additionally, S7 Text contains $t$ tests of differences in reformulation for more unhealthy categories (0.5 to 1.5 and 2.0 to 3.5) compared to the healthiest category (4.0 to 5.0). Overall, these results supported the hypothesis of the increased reformulation for products with lower baseline healthiness.

### Sensitivity analyses

Figures of preexisting trends in reformulation between labelled and unlabelled products are presented in S4 Text. Generally, we did not find differences in reformulation trends before labelling, strengthening the validity of the parallel paths identifying assumption for difference in differences.

Effect sizes were generally marginally larger when we use CEM matching weights in Eq 1 and marginally lower when allowing for differential pretreatment reformulation trends (S5 Text). They were also similar when we combined the 2 approaches. These analyses supported the validity of the difference-in-differences estimates presented above.

## Discussion

In this study, we found that voluntary adoption of the HSR was associated with product reformulation. In NZ, the average HSR rating of foods that adopted HSR increased by 2%, whereas Australian products showed a smaller increase of 1%. These results indicate healthier products post-labelling. The greatest reformulation effects were seen for sodium and sugar contents, whereas energy reductions were modest to negligible. We did not find changes in protein or saturated fat content. Changes in fibre were inconsistent across the 2 countries. However, only 23% and 31% of products in NZ (2019) and Australia (2018), respectively, had adopted HSR. The effect of mandatory labelling may, therefore, not be a linear extrapolation from partial uptake due to the voluntary nature of many FoPL schemes.

Reformulation was generally least for the products with the best baseline nutrient profile (between 4.0 and 5.0 stars) and greatest for products with the worst (0.5 to 1.5 stars). This was consistent with ceiling effects in reformulation, whereby already nutritious products have limited scope for healthier reformulation.

### Comparison with previous studies

This paper contains many improvements over earlier studies. First, it enhances the generalisability of results by studying products adopting HSR over a longer time horizon. Compared to the earlier NZ study [12], it controls for reformulation trends among unlabelled foods. This generally has the effect of reducing the casual impact of HSR compared to a simpler before–after analysis. For instance, our study finds sodium and sugar reductions of −16 mg/100 g and −0.21 g/100 g; the earlier study found larger reductions of −49 mg/100 g and −0.3 g/100 g, respectively. We also became aware of, and corrected for, previous issues with fibre data; some missing values were entered as 0s causing overestimates in a before–after setting. To illustrate, the previous study found fibre increasing by 0.5 g/100 g compared to 0.04 g/100 g for this study. The previous Australian study [13] uses a difference-in-differences design for analysing energy reformulation in 2016, finding a −7.11 kJ/100 g [95% CI: −14.2 to −0.1, $p = 0.04$] reduction, which is close to our estimates (−4.61 kJ/100 g [95% CI: −8.44 to −0.79, $p = 0.006$]). We report upon a larger variety of nutritional outcomes compared to the previous Australian study. The results are lower than the mean estimates (e.g., for sodium, 8.9%, 95% CI: −17.3% to 0.6%, $n = 4$) from a systematic review of industry-led reformulation [7], but this may reflect differences in labelling schemes and study designs in assessed studies. Lastly, we studied

stratification of reformulation effects by baseline healthiness of products and found greater reformulation for unhealthy foods on voluntarily adopting HSR.

FoPL systems include a wide range of designs and policy, and public health authorities are faced with a wide range of effect sizes across various schemes and observation study designs. Our study produces results that are more muted than those arising from the Dutch Choices logo program [9], which highlights positive nutrients. However, this may reflect differences in study design. To the best of our knowledge, no studies have causally assessed the reformulation effect of many widely used or cited FoPL schemes such as traffic light signals highlighting levels of individual nutrients in the United Kingdom [23], the summary-graded Nutri-Score in Europe [24], or warning logos for sodium, energy, sugars, and saturated fats in Chile [25]. Despite the absence of such studies, our results shall serve as a close benchmark for similar graded summary schemes, such as Nutri-Score, which has seen increasing uptake across Europe.

## Strengths and limitations

Our study utilised the most comprehensive NIP datasets from Australia and NZ and used contemporary panel data methods that adjust for all time-invariant, and several time-varying, confounders to estimate the casual reformulation effect of HSR.

Limitations arise from our imputations of missing values. Products that did not display fibre were assumed to have negligible fibre content. Bias in imputing fibre or FVNL imputations in either dataset may have flow-on effects for imputations of HSR ratings. Although the use of fixed effects may ameliorate such bias, it may cause underestimation of reformulation effect on fibre and HSR ratings. Policy-relevant nutrients such as sodium, sugar, energy, and saturated fat were always displayed on NIPs and thus unaffected by such imputations.

An important limitation of our study is that results are not sales weighted. Thus, it is not able to assess changes in overall nutrient consumption that occur because of HSR-caused reformulation. This limitation in the study design was motivated by the fact that sales weights are also affected by HSR—for instance, the demand for unhealthy products may decrease post labelling, which further affects consumption. An analysis of the consumption effects of dietary policy must include both changes in consumer and industry behaviour. This is outside the scope of the study and its datasets, and we aim to address it separately. However, the modest results herein suggest that overall changes to nutrient consumption due to reformulation caused by HSR are likely to be limited. Making HSR mandatory is likely to improve the healthfulness of consumer diets by causing more unhealthy products to adopt the label.

Further, this study focusses on product reformulation effects. Healthier product innovation, where new healthful products enter the marketplace, is another potential effect of HSR. However, the selection of already-healthy products into HSR and the absence of a comparator group for such newly innovated products mean that these innovation effects could not be estimated by this study.

Confounding is a major issue with observational studies, and we, therefore, performed additional sensitivity analyses that match labelled and unlabelled HSR products by baseline product characteristics and undertook analyses controlling for linear differences in reformulation trends between labelled and unlabelled products. Results were robust to these checks (S5 Text).

Further, difference-in-differences estimates rely on the parallel path assumption—reformulation among HSR and non-HSR products would have been comparable, had HSR not been introduced. Although counterfactual, we showed that there are no significant differences in reformulation trends between labelled and unlabelled foods before the adoption of HSR. This strengthened the likelihood of the identifying assumption being valid.

## Implications

HSR, a voluntary FoPL scheme, drives small industry-led reformulation. Such reformulation is greater among less nutritious products, although they are less likely to participate. We note that a major review of the HSR system [26] has identified the need to set high uptake targets (for instance, 70% in 5 years) to maximise the public health impact of the label. Our results imply that policymakers and targets should be mindful of the healthiness of products adopting voluntary labelling schemes. Although market-based mechanisms may also cause near-universal adoption of such schemes in theory [27], such adoption is rarely observed in real life [28] or even the analysis sample used here. Mandatory adoption of FoPL for unhealthy products is likely to maximise the public health gains arising from reformulation and likely also from changes in consumer behaviour.

These reformulation effects are likely to have modest effects on population health. For instance, the previous Australian study, which found an energy reduction of −7.11 kJ/100 g, also estimated the health impact of HSR using simulation models, generating 4,207 disability-adjusted life years (DALYs) averted (95% CI: 2,438 to 6,081; discounted at 3% per annum) for the Australian population in 2010 over the remainder of their lifetime [13]. This equated to roughly 100 minutes of healthy life per person. However, gains were estimated to increase 10-fold if the HSR scheme was made mandatory. As a follow-up to this study, we will be updating estimates on future population health across Australia and NZ.

Although our results show that HSR labelling is associated with reformulation, we did not establish changes in nutrient composition or additives that underpin such reformulation. Such changes, e.g., adding artificial sweeteners, have important consequences for both the health implications [29] and sensory characteristics [30] of the products studied. A lower-level analysis of the changes to ingredients that affect product reformulation and consequences thereof, although beyond the scope of this study, is an important avenue for future research. Irrespective of the density of healthy and unhealthy nutrients in food, there is also a growing literature that highlights the health concerns of consuming ultra-processed foods in general [31,32]. An analysis of policies that improve population diets through increasing consumption of unprocessed and minimally processed foods presents a largely unaddressed vital area for future work.

Another mechanism for the impact of FoPL is through driving healthier consumer purchasing, and there is no consensus on whether such schemes are effective in achieving healthier consumption behaviours [33]. Thus far, the best evidence on HSR is provided by app-based randomised trials, which find no effect of HSR on improving real-world purchasing behaviour, although these results may not be generalisable [2,34]. To enhance the evidence of HSR or comparable graded summary FoPL on affecting consumer behaviour, we are conducting a follow-on study analysing the effect of HSR on purchasing patterns using household panel data.

## Conclusions

In this setting, we found that FoPL schemes such as HSR may play a modest role in driving healthier product reformulation, and such reformulation is higher for the least healthy products. The low uptake of HSR overall, and an even lower rate of labelling for unhealthy products, limits reformulation. To maximise the reformulation effects of FoPL, we suggest that governments should make such schemes mandatory.

## Supporting information

**S1 Text. Descriptive analyses.**
(DOCX)

**S2 Text. Methodological note on reconstituted foods/"as-prepared" NIPs.**
(DOCX)

**S3 Text. Tables underlying Figs 3 and 4.**
(DOCX)

**S4 Text. Check for parallel trends (preexisting differential trends).**
(DOCX)

**S5 Text. Robustness checks (CEM and DDD estimates).**
(DOCX)

**S6 Text. Effects of fibre imputation.**
(DOCX)

**S7 Text. *t* test between most nutritious (4.0 to 5.0) and remaining categories (0.5 to1.5 and 2.0 to 3.5).**
(DOCX)

**S8 Text. STROBE reporting matrix.**
(DOCX)

**S9 Text. Grant extract.**
(DOCX)

## Acknowledgments

We thank Helen Eyles (University of Auckland) for providing the Nutritrack dataset and offering valuable support. We also thank Fraser Taylor (The George Institute for Global Health, Sydney) for his support in providing access to the FoodSwitch dataset.

## Author Contributions

**Conceptualization:** Cliona Ni Mhurchu, Tony Blakely.

**Data curation:** Laxman Bablani, Cliona Ni Mhurchu, Bruce Neal.

**Formal analysis:** Laxman Bablani.

**Funding acquisition:** Cliona Ni Mhurchu, Bruce Neal, Tony Blakely.

**Methodology:** Laxman Bablani, Cliona Ni Mhurchu, Bruce Neal, Christopher L. Skeels, Kevin E. Staub, Tony Blakely.

**Writing – original draft:** Laxman Bablani, Tony Blakely.

**Writing – review & editing:** Laxman Bablani, Cliona Ni Mhurchu, Bruce Neal, Christopher L. Skeels, Kevin E. Staub, Tony Blakely.

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
