## [Editor Report · Decision Letter 0]

11 Mar 2020

Dear Dr Bablani, 

Thank you for submitting your manuscript entitled "The Impact of Voluntary Front of Pack Nutrition Labelling on Packaged Food Reformulation: A difference-in-differences analysis of the Australasian Health Star Rating scheme" for consideration by PLOS Medicine.

Your manuscript has now been evaluated by the PLOS Medicine editorial staff and I am writing to let you know that we would like to send your submission out for external peer review.

Kind regards,

Helen Howard, for Clare Stone PhD 

Acting Editor-in-Chief

PLOS Medicine 

plosmedicine.org

---

## [Decision Letter · Decision Letter 1]

26 Apr 2020

Dear Dr. Bablani,

Thank you very much for submitting your manuscript "The Impact of Voluntary Front of Pack Nutrition Labelling on Packaged Food Reformulation: A difference-in-differences analysis of the Australasian Health Star Rating scheme" (PMEDICINE-D-20-00808R1) for consideration at PLOS Medicine. 

[LINK]

In light of these reviews, I am afraid that we will not be able to accept the manuscript for publication in the journal in its current form, but we would like to consider a revised version that addresses the reviewers' and editors' comments. Obviously we cannot make any decision about publication until we have seen the revised manuscript and your response, and we plan to seek re-review by one or more of the reviewers. 

We expect to receive your revised manuscript by May 15 2020 11:59PM. Please email us (plosmedicine@plos.org) if you have any questions or concerns.

We look forward to receiving your revised manuscript. 

Sincerely,

Emma Veitch, PhD

PLOS Medicine

On behalf of Clare Stone, PhD, Acting Chief Editor,

PLOS Medicine

plosmedicine.org

*We'd ask that you clarify in the paper if the analytical approach reported in the paper was planned out prospectively. Please state this (either way) in the Methods section.

a) If there was a prospective analysis plan (ie from the your funding proposal/protocol) used in designing the study, please include the relevant prospectively written document with your revised manuscript as a Supporting Information file to be published alongside your study, and cite it in the Methods section. A legend for this file should be included at the end of your manuscript. 

b) If there was no such prospective analysis plan, please make sure that the Methods section transparently describes when analyses were planned, and when/why any data-driven changes to analyses took place. 

*In the last sentence of the Abstract Methods and Findings section, please describe the main limitation(s) of the study's methodology.

*In the Limitations section of the Discussion, this mentions the parallel path assumption as being a key issue, which if violated could be a major limitation of the study. Many readers may not know what a parallel path assumption is, and for more general readers (for whom the topic matter may be of great interest, even if they don't understand the specialist methods used here), it would be good to say something briefly about what that assumption is and how/why it is relevant to such methodological approaches as used here.

*I noted that the STROBE reporting tool is used and supplied as a checklist in the supporting files. We'd recommend that you add a brief sentence in the Methods to note that STROBE was used in reporting the study. 

Comments from the reviewers:

Reviewer #1: Bablani and colleagues report the findings of a panel data analysis estimating the impact of voluntary front of package nutrition labelling (health star rating scheme) on package food reformulation from 2013-2019 in Auckland and 2014-2018 in Sydney. The analysis was conducted as a difference-in-difference design with HSR as the exposure variable and change in nutrient composition as outcomes variables (specifically energy, sodium, sat fat, protein, fibre). The authors found that labelled products showed significant decrease in sodium and sugar content in NZ but no/very little change in Australia. My major focus was on the robustness of the choice of the comparator in this analysis. In these types of panel data analyses setting up a natural experiment type design, the choice of the control is key. I think the authors realized this in their analysis with a large part of their focus on testing the strength of the assumption. In addition to controlling for confounding within the regression models, the authors were able to examine whether changes showed consistency prior to labelling for both labelled and unlabeled groups and also included three additional robustness checks: (1) CEM, a matching technique that balances nutrient information between both groups (2) controlled for a pre-existing linear trend (3) with finally a combination of the all approaches. The thinking here is that they have tried to design this natural experiment analysis to minimize confounding as similar as possible to randomized controlled trial principles to allow for causal inference framework. Due the substantial thought and efforts given towards the design and analysis, I felt the methodology was very sound. I am not a content expert in this area, so I couldn’t comment on the findings implications for Australia and New Zealand public health policy. If I were however to nitpick, I wonder how generalizable the findings are to other countries?

Reviewer #2: The manuscript deals with a relevant topic: the effect of FOPL on product reformulation. Although the manuscript is well-written and data correctly analyzed, there are several major issues that should be addressed before acceptance. Please find detailed comments below.

Introduction

- An explanation of how FOPL is expected to encourage reformulation is needed between the 1st and the 2nd paragraph

- The authors should include references to the effects of the Chilean food law on food reformulation.

Materials and Methods

- The databases should be more clearly explained, i.e. what type of information is available in the database. The authors should better explain how comparable the NZ and Australian data are.

Results

- For the products that were reformulated, do the authors have information about how they were reformulated? I.e., was sugar reduction achieved by adding sweeteners? was salt-reduction achieved by adding KCl? This information is highly relevant to fully understand the potential effects of reformulation on intake of additives and also on changes in the sensory characteristics of products that could encourage changes in food preferences. This should also be included in the discussion.

- Can the authors separate product reformulation and product innovation (i.e. new healthful products entering the marketplace)? If not, they should discuss this as a limitation and additional potential effect of FOPL.

Discussion

- The discussion should include a critical overview of the HSR given that it is mainly used in the more healthful products. The authors state that results suggest that the use of HSR should be incentivized. However, results from the study suggest that it should be made compulsory

- The authors should include an in-depth discussion of the limitations of product reformulation in the light of the growing body of evidence associating ultra-processed products with NCDs, regardless of their nutritional composition.

Reviewer #3: The manuscript "The Impact of Voluntary Front of Pack Nutrition Labelling on Packaged Food Reformulation: A difference-in-differences analysis of the Australasian Health Star Rating scheme" reports changes in nutritional quality of packaged foods and beverages from Australia and New Zealand after the implementation of the Health Star Rating (HSR) scheme, comparing such changes between products adopting the front-of-package label (FOPL) and the ones which did not adopt it. The report shows that foods using the FOPL had a small improvement in the HSR score and some specific nutrients (sugars, sodium and fiber, depending on the country); reformulation variated depending on the baseline healthiness of the products, being greater among those products with a worst baseline HSR score. 

GENERAL COMMENTS

This is a well written manuscript, easy to follow by expert and non-expert readers. The current analysis includes 2 longitudinal samples (one per country) and adds to the existent reports on the impact of HSR by including larger samples of products, and considering the counterfactual of products with no FOPL use, which controls by potential changes in food formulation given to other factors. However, it has important weaknesses that authors should overcome. 

The report seems to be focused importantly on readers from Australia or New Zealand who are well aware of the system. This is evident at different levels. In order to properly reach a broader spectrum of readers around the world, I suggest to present a picture of the FOPL (it may be added as supplementary material). Also, it should be indicated the overall usage of the FOPL in each country earlier in the text, currently this information is only reported at the end of the manuscript. 

In the same line, the report should aim to inform local health authorities and readers, but also be helpful to other readers form other countries who might be discussing which FOPL schemes to implement. In this regard, the discussion should expand the comparison of the reported results with the ones reported with other FOPL schemes. More importantly, in order to increase the relevance of the results, I strongly suggest including in the analyses the overall reformulation of the analytical sample. For example, how meaningful is the decrease in 42 mg of sodium per 100g/ml for labeled foods in the context of the partial use of the label? How much has the overall food supply (or the analytical sample) improved? That information is surely more helpful for informing decisions (for instance the need to implement mandatory labels) both locally and in other countries (even understanding that a difference-in-difference analysis would not be possible). 

SPECIFIC COMMENTS

Methods

The sample could be better described, specifying food groups considered, and the relative proportion of them included in each group (labeled vs unlabeled products). I suppose authors did not provide results by food group given the magnitude of the results. However, understanding the food group composition of the sample and specially of the labeled and unlabeled groups is key to better understand and interpret the results. Differences in the relative proportion of food groups may be influencing some observations. I understand that the sensitivity analyses balancing groups by pre-labelling nutrient information is aiming to address this issue, but matching by nutrient characteristics is not the same that matching by food groups. Reformulation may be easier in a given food group due to technological characteristics of that particular food group. On the other hand, even if the results are not sale-weighted, authors should provide more information about how relevant are the products included (any information on either market or dietary share is needed). Currently, author provides information about the relevance of supermarket, but something similar should made available at the level of products. 

How was handled the nutrition information of products needing reconstitution (i.e. powder or concentrated soups or fruit drinks)?

I understand authors decided to use the imputed scores and I agree with the decision, but I suggest to include a sensitivity analysis grouping products based on the actual use of the FOPL, assessing just the nutrient content and energy density (given the HSR score will not be possible to evaluate). 

Sensitivity analyses aim to validate in different way the counterfactual; one of such considers the reformulation trend before the implementation of the measure. How was that possible in the case of Australia, where only one time point was collected before the implementation?

Results

In order to better understand the dynamics of the HSR implementation, I suggest adding a graph showing the percentage of foods using the label every year (I imagine the current use was achieved after several years of implementation).

Table 1 and 2 display symbols to represent p-values, however such symbols are not used in the tables. 

Discussion

Besides what said under 'General Comments', authors could further discuss on the implications of the results. For instance, they quote a previous report modeling the impact of the reported reformulation on DALYs, without indicating the extend of the reformulation reported in that specific study. Moreover, regarding food technology and processing of food, further discussion could be done on the addition of fiber to food, ingredients that could replace sugars or sodium on foods and beverages, or the concern regarding processing itself. When expanding the discussion to the effect of other FOPL schemes on reformulation, differences between such schemes could be also discussed (the ones focused on positive aspects vs the ones focusing in negative aspects, etc).

[LINK]

---

## [Decision Letter · Decision Letter 2]

27 Jun 2020

Dear Dr. Bablani,

Thank you very much for submitting your revised manuscript "The Impact of Voluntary Front of Pack Nutrition Labelling on Packaged Food Reformulation: A difference-in-differences analysis of the Australasian Health Star Rating scheme" (PMEDICINE-D-20-00808R2) for consideration at PLOS Medicine. 

Your paper was evaluated by a senior editor and discussed among all the editors here. It was also sent to two of the original reviewers. The reviews are appended at the bottom of this email and any accompanying reviewer attachments can be seen via the link below:

[LINK]

In light of these reviews, I am afraid that we still will not be able to accept the manuscript for publication in the journal in its current form, but we would like to consider a further revised version that addresses the reviewers' and editors' comments. Obviously we cannot make any decision about publication until we have seen the revised manuscript and your response, and we plan to seek re-review by one or more of the reviewers. 

We expect to receive your revised manuscript by Jul 20 2020 11:59PM. Please email us (plosmedicine@plos.org) if you have any questions or concerns.

We look forward to receiving your revised manuscript. 

Sincerely,

Thomas McBride, PhD

Senior Editor 

PLOS Medicine

plosmedicine.org

1- Thank you for providing your STROBE checklist. Please replace the page numbers with paragraph numbers per section (e.g. "Methods, paragraph 1"), since the page numbers of the final published paper may be different from the page numbers in the current manuscript.

2- Thank you for providing your grant application as a supplementary text. I notice you redacted portions, and the pages are labeled “IN CONFIDENCE”. Please confirm that you are comfortable with this document being published alongside your manuscript. 

3- Additionally, please note in the methods section any analyses that differ from those that were planned, and provide transparent explanations for differences that affect the reliability of the study's results. For example, if a reported analysis was performed in response to a reviewer’s request, please note this. If an analysis was based on an interesting but unanticipated pattern in the data, please be clear that the analysis was data-driven. If hypotheses that were not included in the original study design later became important to test because new evidence became available from other studies, please explain the situation, so that it is clear whether new analyses were data driven or added for another reason. 

4- In the Abstract Conclusions, please address the study implications without overreaching what can be concluded from the data; the phrase "In this study, we observed ..." and “Our results suggest…” may be useful.

5- Please edit the Discussion Conclusions similarly. “*In this setting, we found that* FoPL schemes such as HSR *may* play…” and “To maximise the reformulation effects of FoPL, *we suggest* governments make such schemes mandatory.” would be appropriate.

6- Throughout the manuscript, where relevant, please include p-values alongside 95% CIs

Comments from the reviewers:

Reviewer #2: no further comments

Reviewer #3: Authors have provided more details about the policy and the sample of foods, which has improved importantly the clarity of the current version of the manuscript. However, there are some important aspects to consider in this revised version. 

The abstract state: "We studied the impact of voluntary adoption of HSR on food reformulation overall, and for more- versus less-healthy foods". To me, this means the manuscript will report the reformulation of all available foods after the voluntary adoption of HSR, which is not the case given it reports the reformulation of labeled foods compared to unlabeled foods. As commented in my previous review, I do think it is important to additionally present data on the changes of food composition for the overall food supply collected in both countries, even with no counterfactual, a simple pre-post analysis, to have a sense of the overall impact. Author's response to that comment is that results are not sales-weighted. Sales-weighted data would allow providing a greater relevance to reformulation of foods that have a greater market share, compared to reformulation of foods that have little participation in the market. I agree this is very relevant and therefore a limitation of the study. However, sales-weighted data are not needed to evaluate the impact of reformulation on the food supply (i.e., for the purpose of this study, non-seasonal packaged foods available at the main supermarkets during the years of data collection). If authors decline to include this extra analysis, please remove the sentence of the abstract, which I think may be confusing for other readers that, as me, could expect to see actual data on the impact of HSR on food reformulation overall.

Please revise the following statement from the abstract states: "A limitation of our study is that results are not sales-weighted. Thus, it is not able to assess changes in the overall food supply that occur because of HSR-caused reformulation." As previously commented, the fact that results are not sales-weighted does not mean that authors cannot assess changes in the overall food supply. I might be missing something, in that case please provide an explanation for other readers as me who do not see the connection. 

In author summary, please consider rephrasing this statement "Initially unhealthy products increase their HSR rating by more than 0.1 stars, while healthier products show less reformulation - a 1 star increase in initial healthiness reduces reformulation by around 0.04 stars."

Regarding the use of HSR label, the introduction states: "Since its introduction, HSR has seen reasonable acceptance, and was displayed on about 23% of NZ products in 2019, and 31% of Australian products in 2018 (Appendix S1 graphs the percentage of foods using HSR across years in Australia and NZ)." Can you please rephrase clearly indicating the percentage of use has been continuously increasing since the adoption of the policy until reaching those percentages in 2019 (to better reflect what is seen in S1 graphs). Moreover, from table 1 one could interpret the HSR label adoption was about 6% in New Zealand (1785 unique products out of 28053) and 8.5% in Australia (2462 unique products out of 26605). Please provide an explanation for those differences? Are they explained by the gradual adoption of the label? Was there a lower adoption rate within the sample you collected?

Another concern I have is the different composition (regarding food groups) of the labeled subset and the counterfactual unlabeled subset. As shown in S1 table, compared to never labeled products, cereal products are 2 times more frequent within the subset of foods adopting the label (8.8 vs 16.7%, respectively), whereas sugars and related products are 4 times less present in that group (2.5% vs 0.6%, respectively). Those different food group distributions may be explained by differences in the technological feasibility for reformulation between food groups. Thus, I think a bold sensitivity analysis would be something similar to the first one considered (i.e., coarsened exact matching (CEM) is a non-parametric matching technique that balances pre-labelling nutrient information between HSR products and products that never received HSR labelling), but balancing pre-labelling food groups classification between HSR products and products that never received HSR. Moreover, I suggest S1 table displays percentages of food groups considering the subset (i.e., never HSR labelled vs adopted HSR) as denominator, in order to clearly see the difference in food groups composition between both subsets.

Results section: Values for sodium and sugars in the text are different than the ones displayed in Table 2.

First paragraph of the discussion should include the prevalence of labelled products in order to better interpret the magnitude of those changes on the overall food supply (or provide information on the extra analysis proposed, if considered).

Regarding the limitation that results are not sales-weighted, and thus authors are not able to assess changes in the overall food sample that occur because of HSR-caused reformulation, please considered my earlier comment made regarding the same issue 

Although I understand why the sentence "There is also a growing literature that highlights the health concerns of consuming ultra-processed foods in general" was added, please consider including "despite the content of healthy and unhealthy nutrients of such foods" or something similar. As it is right now, I do not think the idea will be clear for all readers. 

"NIP" is defined several times within the text.

S2: Please describe what Nutrition Information Panel (NIP) stand for earlier in the text.

S3: Title reads "S3 Appendix: Tables underlying Figure 4,5" I imagine it should read "S3 Appendix: Tables underlying Figure 4". Please be consistent with headings and order of columns between S3 and tables 2 and 3.

[LINK]

---

## [Decision Letter · Decision Letter 3]

30 Sep 2020

Dear Dr. Bablani,

Thank you very much for re-submitting your manuscript "The Impact of Voluntary Front of Pack Nutrition Labelling on Packaged Food Reformulation: A difference-in-differences analysis of the Australasian Health Star Rating scheme" (PMEDICINE-D-20-00808R3) for review by PLOS Medicine.

I have discussed the paper with my colleagues and the academic editor and it was also seen again by one of the original reviewers. I am pleased to say that provided the remaining editorial and production issues are dealt with we are planning to accept the paper for publication in the journal.

[LINK]

We look forward to receiving the revised manuscript by Oct 07 2020 11:59PM. 

Sincerely,

Thomas McBride, PhD

Senior Editor 

PLOS Medicine

plosmedicine.org

Requests from Editors:

1- Please edit the data statement to read:

“Because of commercial and legal restrictions to the use of copyrighted material it is not possible to share data openly which reveal the product or supermarket names, but unredacted versions of the dataset are available with a licensed agreement that they will be restricted to non-commercial use. For access to Nutritrack, please contact the The National Institute for Health Innovation at the University of Auckland at enquiries@nihi.auckland.ac.nz. For access to FoodSwitch, please contact Fraser Taylor, managing director for Foodswitch ftaylor@georgeinstitute.org.au or foodswitch@georgeinstitute.org.au.”

2- You use both “less healthy” and “unhealthy” foods, please be consistent throughout the manuscript.

3- Early in the Abstract Methods and Findings, please note the number of products in the analysis and specify the classes of product included (cereals, etc).

4- Throughout the manuscript, please check that you are consistent with your use of positive and negative changes when describing declines or increases in content. For example, the Abstract Methods describes: “ Labelled products showed a -4.0% [95% CI -6.4% to -1.7%, p=0.001] relative decline in sodium content in NZ, and there was a 1.4% [95% CI: 2.7% to 0.0%, p=0.045] sodium change in Australia.”, when it seems that both countries experienced a decrease (i.e., should it be -1.4%?).

5- Additionally, please be clear that the difference in sugar content observed in Australia was not significant (you do so in the Author Summary, but not the Abstract).

6- The Abstract Methods and Findings could note that you did not find changes in protein or saturated fat content.

7- As this study is observational, please avoid language that implies causality (e.g., “impact”, “effect”) throughout. Examples include the last sentence of the Abstract Background, the first sentence of the Abstract Conclusions, the third point of the Author Summary, the end of the Introduction, the opening sentence of the Discussion.

8- It may be also worth noting the limitations of the observational design alongside other limitations in the Abstract.

9- The Introduction could include a bit more on the health impacts of healthier diets (e.g., major diseases linked to sodium, sugar, and fiber content). 

10- Thank you for providing p values. Please remove the italics.

11- Please also remove the italicized text used for emphasis (e.g., bottom of page 13 “The second row contains the *change* in reformulation when baseline HSR *increases* by 1. ”

12- Bottom of page 13: stray equals sign? (=0.17-0.04)

13- The Funding and Competing interests statements can be removed from the main text and reside in the metadata fields.

14- Please move the Acknowledgements section to the end of the main text.

15- Thank you for providing the redacted grant text. We note that this file has a copyright symbol on it. Please excerpt the non-redacted text into a new document and replace S9, noting what has been done in the Supplementary file legend.

Comments from Reviewers:

Reviewer #3: I have no further comments.

[LINK]

---

## [Editor Report · Decision Letter 4]

19 Oct 2020

Dear Dr Bablani, 

On behalf of my colleagues and the academic editor, Dr. Marcela Reyes, I am delighted to inform you that your manuscript entitled "The Impact of Voluntary Front of Pack Nutrition Labelling on Packaged Food Reformulation: A difference-in-differences analysis of the Australasian Health Star Rating scheme" (PMEDICINE-D-20-00808R4) has been accepted for publication in PLOS Medicine. 

PRODUCTION PROCESS

Before publication you will see the copyedited word document (within 5 business days) and a PDF proof shortly after that. The copyeditor will be in touch shortly before sending you the copyedited Word document. We will make some revisions at copyediting stage to conform to our general style, and for clarification. When you receive this version you should check and revise it very carefully, including figures, tables, references, and supporting information, because corrections at the next stage (proofs) will be strictly limited to (1) errors in author names or affiliations, (2) errors of scientific fact that would cause misunderstandings to readers, and (3) printer's (introduced) errors. Please return the copyedited file within 2 business days in order to ensure timely delivery of the PDF proof. 

If you are likely to be away when either this document or the proof is sent, please ensure we have contact information of a second person, as we will need you to respond quickly at each point. Given the disruptions resulting from the ongoing COVID-19 pandemic, there may be delays in the production process. We apologise in advance for any inconvenience caused and will do our best to minimize impact as far as possible.

PRESS

PROFILE INFORMATION

Thank you again for submitting the manuscript to PLOS Medicine. We look forward to publishing it. 

Best wishes, 

Thomas McBride, PhD

Senior Editor 

PLOS Medicine

plosmedicine.org